# Do the Dietary Intakes of Pregnant Women Attending Public Hospital Antenatal Clinics Align with Australian Guide to Healthy Eating Recommendations?

**DOI:** 10.3390/nu12082438

**Published:** 2020-08-13

**Authors:** Kaylee Slater, Megan E. Rollo, Zoe Szewczyk, Lee Ashton, Tracy Schumacher, Clare Collins

**Affiliations:** 1School of Health Sciences, Faculty of Health and Medicine, The University of Newcastle, Callaghan, NSW 2308, Australia; Kaylee.Slater@uon.edu.au (K.S.); megan.rollo@newcastle.edu.au (M.E.R.); Zoe.Szewczyk@hmri.org.au (Z.S.); lee.ashton@newcastle.edu.au (L.A.); 2Priority Research Centre in Physical Activity and Nutrition, The University of Newcastle, Callaghan, NSW 2308, Australia; tracy.schumacher@newcastle.edu.au; 3Hunter Medical Research Institute (HMRI) Lot 1, Kookaburra Circuit, New Lambton Heights, NSW 2305, Australia; 4Department of Rural Health, The University of Newcastle, Tamworth, NSW 2340, Australia; 5Priority Research Centre for Health Behaviours, The University of Newcastle, Callaghan, NSW 2308, Australia

**Keywords:** pregnancy, nutrition, dietary intake, dietary guidelines, food-based guidelines

## Abstract

The maternal diet influences the long-term health status of both mother and offspring. The current study aimed to compare dietary intakes of pregnant women compared to food and nutrient recommendations in the Australian Guide to Healthy Eating (AGHE) and Nutrient Reference Values (NRVs). Usual dietary intake was assessed in a sample of women in their 3rd trimester of pregnancy attending antenatal outpatient clinics at John Hunter Hospital, Newcastle, New South Wales (NSW). Dietary intake was measured using the Australian Eating Survey, a validated, semi-quantitative 120-item food frequency questionnaire. Daily food group servings and nutrient intakes were compared to AGHE and NRV targets. Of 534 women participating, none met the AGHE recommendations for all food groups. Highest adherence was for fruit serves (38%), and lowest for breads and cereals (0.6%). Only four women met the pregnancy NRVs for folate, iron, calcium, zinc and fibre from food alone. Current dietary intakes of Australian women during pregnancy do not align with national nutrition guidelines. This highlights the importance of routine vitamin and mineral supplementation during pregnancy, as intakes from diet alone may commonly be inadequate. Future revisions of dietary guidelines and pregnancy nutrition recommendations should consider current dietary patterns. Pregnant women currently need more support to optimise food and nutrient intakes.

## 1. Introduction

Dietary intakes during pregnancy influence long-term health of both the mother and offspring [1]. Maternal diets can either enhance or compromise the mother’s health status during pregnancy, impacting foetal development and influencing both maternal and offspring risk for non-communicable diseases later in life [2,3,4,5]. A woman with excessive gestational weight gain is more likely to develop hypertensive disorders and diabetes post pregnancy and transgenerational obesity in the offspring [5]. Maternal diet quality and adequacy of nutrient intake is associated with foetal development [6].

Maternal diets are often characterised by high intakes of energy-dense and nutrient-poor (EDNP) foods and are high in energy, saturated fat, added sugars and sodium and low in dietary fibre [1,7]. These dietary patterns may impact on adequacy of specific micronutrient intakes such as folate, iron, calcium and zinc, which are important in optimising reproductive health as well as foetal growth and development [1,8]. It has been reported in two systematic reviews that in high income countries the maternal diet does not align with life-stage specific national recommendations, increasing the likelihood of suboptimal micronutrient and macronutrient intakes [9,10]. Australian cross-sectional studies, all with over 600 participants, suggest that dietary intakes of pregnant women are likely to be inadequate and not meeting recommendations for macronutrients and food groups, with the exception of intakes of fruit [1,7,11,12,13,14]. Interestingly, research suggests that there are small differences between the diets of pregnant and non-pregnant women [1,15]. However, Savard et al. suggested that pregnant women have greater diet quality overall, raising the concern for overall diet quality amongst women of childbearing age [16].

Dietary guidelines, food selection guides and healthy eating messages form a core component of worldwide strategies to prevent non-communicable disease and encourage consumption of a wide variety of foods [14,17,18,19]. Pregnancy specific dietary guidelines have been developed, such as those included in the Australian Guide to Healthy Eating (AGHE) [18]. Pregnancy nutrition guidelines provide evidence-based recommendations for optimal maternal nutrition intake to promote maternal and infant health [14,18,20]. The AGHE describes types and amounts of foods that pregnant women should be consuming in order to meet macro- and micronutrient intakes [18].

Food group intakes recommended in the AGHE were derived by linear programming, where a modelling approach was used to create age and sex-specific “Foundation Diets” [18]. These Foundation Diets were determined using a range of combinations of amounts and types of foods that would meet all the nutrient reference values (NRVs), with suitable energy requirements of the smallest and least active people for each sub-group [18]. “Total diets” were subsequently derived from Foundation Diets, with additional food options to meet energy and nutrient needs for taller and more active individuals [18].

Current data reporting on the adequacy of maternal dietary intakes relevant to current national dietary guidelines remains limited. Studies that have compared dietary intake data to the most recent 2013 AGHE indicate that pregnant women are not meeting dietary recommendations, with less than 2% meeting daily vegetable serve recommendations, 1% meeting the grains recommendations and only 1% achieving higher than four serves of dairy per day [7,12,14,21]. However, these studies have largely used dietary intake data collected prior to the AGHE update in 2013 [1,7,12,14]. As such, there is a need to compare current eating patterns of pregnant women to the most recent 2013 AGHE. In doing this, future dietary guidelines can use current data in modelling approaches.

Therefore, the aim of the current study was to: (i) evaluate whether dietary patterns align with current AGHE food group recommendations for pregnancy in a contemporary sample of Australian pregnant women and (ii) describe food group intakes in the sub-group who meet NRVs for specific micronutrients (folate, iron, calcium, zinc and fibre) important in pregnancy.

## 2. Materials and Methods

### 2.1. Study Design and Participants

The current study is a primary data analysis of an observational study. Participants were pregnant women attending the public antenatal outpatient clinic and planning to deliver at John Hunter Hospital in Newcastle, NSW, Australia. Women were eligible to participate if they were aged over 18 years, between 28 and 36 weeks of gestation, and proficient in English. Participants were not excluded based on illnesses or known medical conditions. Ethics approval was received from the Hunter New England Human Research Ethics Committee (approval number 16/07/20/4.07) and University of Newcastle Human Research Ethics Committee (H-2017-0101).

### 2.2. Recruitment

Recruitment methods included a media release, social media posts on the Hunter Medical Research Institute and University of Newcastle Facebook pages, and flyers with pull-off tabs located in the pathology department and John Hunter Hospital antenatal clinic waiting rooms. Subsequently, the majority of the recruitment came through direct contact from trained research assistants, with pregnant women approached in the John Hunter Hospital Antenatal clinic waiting room. One out of five of the days at the John Hunter Hospital Antenatal clinic was a high-risk Diabetes clinic. Those interested in participating were screened for eligibility, with those eligible then choosing to provide consent to participate. Women who were interested but not yet 28 weeks’ gestation were invited to provide a name and email address for the survey to be emailed when they reached 28 weeks’ gestation. Those unable to complete the survey during their attendance or choosing not to complete it in the waiting room were sent a reminder email, together with either the incomplete, or partially completed survey. Data were collected, using REDcap software (Vanderbilt University, Nashville, TN, USA, version 8.11.3) [22] on iPad devices during a convenience sampling period from March 2018 to November 2018.

### 2.3. Data Collection

The 165-question survey comprised two parts: (1) general and maternal health and (2) dietary intake assessment.

#### 2.3.1. General Demographics and Maternal Characteristics

Four sections, totalling 30 questions, collected demographic data including age, education level, nationality, marital status and income along with information related to maternal health for past (if relevant) and current pregnancies, such as weight gain, medications, smoking status and past and current pregnancy healthcare. Questions were in both multiple choice and short answer format and were created to be comparable to questions asked in the Australian Longitudinal Study of Women’s Health (ALSWH) and the Women and their Children’s Health study. The survey questions are provided in Appendix A.

#### 2.3.2. Dietary Intake Assessment

Usual food and nutrient intakes over the previous 3–6 months were assessed using the Australian Eating Survey (AES), a validated self-administered, semi-quantitative food frequency questionnaire (FFQ), comprising of 120 food items [23]. Additional to the AES, there was a 15-question survey that contained questions relating to age, height, weight and behavioural aspects of eating and included food items such as soft serve and salads that are relevant to pregnancy due to risk of listeria exposure. Answers were assessed using a Likert Scale with response options ranging from never to ’7 times per day’. The FFQ includes a comprehensive list of foods, including drinks, milk and dairy foods, breads and cereals, sweet and savoury snacks, main meals, other foods, vegetables and fruit. The AES is used to estimate usual dietary intakes of Australian adults and has been assessed for comparative validity relative to weighed food records and for fruit and vegetable intakes using plasma carotenoids [23,24]. Standard portion sizes were determined for each AES item in the survey, using data from the most recent National Nutrition Survey [25]. An example of a standard portion size of an item is a slice of bread [23]. The food and beverage weight per serving, used in the calculation of food group servings (as serves per day) is consistent with sizes specified in the AGHE (Table 2). Nutrient intakes from the AES FFQ were computed using data in the AUSNUT 2011–13 database [26].

### 2.4. Australian Guide to Healthy Eating (AGHE)

The AGHE is Australia’s current national food selection guide. It was designed for healthy individuals, including pregnant women, and those with common health conditions but not for specific medical conditions or the frail elderly [18,27]. The AGHE encourages daily food choices from each of the five core food groups: (i) grain (cereal) foods; (ii) lean meats and poultry, fish, eggs, tofu, nuts and seeds and legumes/beans; (iii) vegetables and legumes/beans; (iv) fruit and (v) milk, yogurt, cheese and/or alternatives [18]. Additionally, the energy dense and/or nutrient poor “extras” or discretionary choices group contains foods that are not necessary for a healthy diet and only recommended for consumption in limited amounts, depending on total energy needs [27]. Women’s food group intakes were compared to the AGHE food group servings. Women were said to meet a food group if their intake either met or exceeded the AGHE values, except for the “extras” category, which was reported as the percentage of total energy derived from AGHE core and discretionary food groups (Table 2).

### 2.5. Nutrient Reference Values (NRVs)

NRVs are specific daily nutrient intake targets developed by the National Health and Medical Research Council of Australia, associated with better health outcomes and lower risk of nutritional deficiencies [28]. The estimated average requirement (EAR), adequate intake (AI) and acceptable macronutrient distribution range (AMDR) are the most appropriate comparison values for population intakes. The EAR describes the daily nutrient target that should meet the requirements of half the healthy population at any particular life-stage and gender group [28]. An AI is used when an EAR is unable to be set and describes the average daily nutrient level that is assumed to be adequate by a group (or groups) of apparently heathy people [28]. AMDRs are recommended ranges for the percentage of daily energy intake from macronutrients [28]. Nutrient values for each participant were compared to the NRVs to determine whether pregnant women were meeting or not meeting the NRVs.

### 2.6. Statistical Analysis

To improve the validity of the study, energy intake mis-reporting was explored using cut-offs recommended by Meltzer et al. (2008), excluding those who reported daily energy intakes <4.5 or >20.0 MJ/d [29]. Descriptive statistics were used to summarise demographic characteristics, including age, education level, nationality, income, marital status and smoking status across the women. The main outcome measures of the study were daily servings of food groups (serves/day) and proportions of women meeting the AGHE recommendations and NRVs. Daily food group intakes in servings were calculated using the AGHE and compared with the AGHE recommendations for pregnant women aged 19–50 years. To determine the proportion of women achieving adequate intakes, macronutrients and micronutrients were compared with pregnancy specific values (EARs, AIs and AMDRs where applicable). The median daily food servings and macronutrient intakes for the subgroup who met the NRVs for calcium, zinc, iron, folate and fibre were reported. Data were tested for normality, with normally distributed data reported as mean [95% confidence interval (95% CI)] and non-normal data as the median [interquartile range (IQR)]. All data manipulation and analyses were performed using SPSS, version 22 (IBM Corp., Armonk, NY, USA).

## 3. Results

Of the 1115 women who expressed interest in participating and started the online screener, 169 were ineligible or did not complete screening. Of the remaining 946 women, six did not provide consent. For inclusion in the current analysis, participants needed a complete response to all 120 questions in the AES food list, resulting in a final sample of 534 women (Appendix A). Demographic characteristics of the women with complete data are summarised in Table 1. Women self-reported their pre-pregnancy height and weight, which was used to calculate BMI and categorise their weight status. These values were also measured by midwives at the clinic for accuracy of reporting. A total of 88 women did not report either weight and/or height. Pre-pregnancy BMI was calculated for 446 women, with 57.1% classified as overweight or obese, which is consistent with the general trend amongst women in Australia (total 59.8% overweight or obese) [30]. Where supplement use was reported, types of supplements and medication were not consistently reported and therefore excluded from this study. Additional questions related to food intolerance and food allergies were not asked as it was deemed unnecessary for the overall aim of the research.

Daily food consumption is summarised in Table 2. Food group intakes in daily food group servings and nutrient intakes are reported for both the total sample of women (*n* = 534) and for the sub-group least likely to have misreported total energy intake (*n* = 503). Of the 31 women identified as mis-reporters, 28 reported a total energy intake less than 4.5 MJ/day, and three reported a total energy intake over 20.0 MJ/day. The median and interquartile range (IQR) for percentage energy attributed to nutrient-dense core foods [67(58–75)] and energy-dense, nutrient-poor noncore foods [33(25–42)] are reported in the table. Fats, oils and discretionary items were included in nutrient-poor noncore foods and were not reported as an additional food group due to the set-up of the FFQ.

The percentage of women achieving the daily food group recommendations according to the AGHE is summarised in Table 3. There were no women who achieved AGHE food group servings for all five food groups. Fruit was met by the largest number of women (*n* = 204, 38.2%), whereas breads and cereals were met by the least (*n* = 3). Legumes, which are considered both a vegetable and meat alternative, were added only to the vegetables group for the purpose of this study. Eggs, tofu and nuts were considered as part of the meat and alternatives food group.

The percentage of pregnant women meeting NRVs important in pregnancy are reported in Table 4. The AMDR for protein (*n* = 455, 85.2%), EAR for vitamin C (*n* = 525, 98.5%) and EAR for phosphorus (*n* = 516, 96.6%) was met by the largest number of women, whereas 39 (7.3%) women achieved less than 10% daily energy from saturated fat. Sodium had the lowest rate of adherence, with *n* = 32 (6%) within the AI of 460–920 mg. The number of women exceeding 920 mg/day of sodium was 493 (94.4%), and 3 women (0.6%) consumed under 460 mg/day. This analysis was re-run by BMI categories, using the 446 participants who self-reported height and weight. This data can be viewed in Appendix A.

Median (IQR) food group intakes (servings/day) for those who achieved the pregnancy NRVs for folate, calcium, zinc, fibre and iron were 7.5 (5.2–8.3) (breads and cereals), 2.3 (1.2–3.4) (fruit), 5.5 (2.4–8.0) (vegetables), 0.9–2.1 (1.5–2.7) (dairy), and 4.3 (1.7–9.2) (meat and alternatives). Out of 534 women, four women met all five pregnancy NRVs; however, one of these women misreported. Out of the three women who met the NRVs and were not classified as mis-reporters, one did not meet any of the AGHE food group targets; one only met the dairy food group (3.1 serves/day); and one met the food group intake for meat and alternatives (6.7 serves/day), vegetables (7.1 serves/day), fruit (3.3 serves/day) and breads and cereals (8.79 serves/day). None of the three women met all of the AGHE food group targets for pregnancy. Only four women met the NRVs for all five nutrients important in pregnancy; therefore, little value can be gained from further analysis of this aim.

## 4. Discussion

The principal aim of the current analysis was to evaluate whether Australian pregnant women were eating in accordance with the current AGHE and to report food group intakes of the subgroup of women who met the NRVs for folate, calcium, iron, zinc and fibre, from food intake alone. Results indicate that pregnant women may not be able to meet AGHE recommendations for all food groups nor achieve national recommendations for key micronutrients from food intake alone. The few women (*n* = 4) who met all five of the key pregnancy NRVs had food group intakes that differed to those recommended in the AGHE; however, one of these women classified as a mis-reporter. Pregnancy is a time where women have increased motivation to make dietary improvements [31]. Despite this motivation, within a contemporary sample of pregnant women, they are still not meeting the recommendations. It is also important to acknowledge the large increases in food group recommendations between pregnant and non-pregnant women. Additionally, pregnant women are at increased risk of digestive complications such as nausea and vomiting [32], constipation [33] and reflux [34], potentially hindering their ability to meet the large amount of food recommended in the AGHE.

Nationally representative Australian studies have compared the dietary patterns of pregnant women to both previous and current AGHE versions. These dietary patterns were primarily drawn from studies published between 2011–2018, with their data mostly collected prior to 2013 [1,7,12,14]. This study uses current eating patterns, collected in 2018 for comparison to the most recent 2013 AGHE. Australian dietary patterns have evolved over time as foods available also change, necessitating more current data to compare to national recommendations [35].

Similar to previous studies, the current analysis demonstrates that pregnant women are not meeting nutrition guidelines [1,7,12,14]. More women met the guideline for fruit than any other food group. In a cross-sectional study by Blumfield et al., an FFQ was used to assess intakes in a cohort of 606 pregnant women from the ALSWH [1]. With data collected in 2003, the median fruit intake was 2.2 serves per day, with 55.4% meeting the recommendations. This compares to the current study where the median intake was 1.71 serves of fruit per day, with only 38.2% meeting the target of two serves per day (*n* = 204) [1]. On the other hand, vegetable intake was greater compared to previous studies, with 26.6% of women from the current study meeting the guideline of five serves per day, which may have been an artefact of using a different FFQ compared to the other studies. Additionally, using an FFQ as a measure of dietary assessment may miss food items that are not included in the food list, or food items that are consumed in larger portion sizes than those in the FFQ. Australian cross-sectional studies by Lee et al. (*n* = 1570) and Malek et al. (*n* = 857) saw 10% and 10.3% meeting five serves of vegetables per day, respectively [7,12]. According to the AGHE, legumes are considered part of both the vegetables and meat alternatives food groups [18]. The current study analysed legumes as part of the vegetable food group, potentially contributing to the greater adherence to the AGHE recommendation for vegetables, than seen in previous studies.

Interestingly, a small number of women met the dairy and meat/vegetarian alternatives food groups, 13.5% (*n* = 72) and 18.2% (*n* = 97), respectively. In contrast, a cross-sectional study by Forbes et al. demonstrated that 50% of women increase their milk consumption during pregnancy but reduce their intake of meat, contrary to recommendations [36]. A possible reason for this finding may be that pregnant women commonly consume dairy foods, specifically cow’s milk, to relieve heartburn, a common symptom reported amongst pregnant women [37]. Additionally, a cross-sectional study of 148 pregnant and 130 non-pregnant women reported that pregnant women consumed larger amounts of dairy and beef than non-pregnant women [15]. If only a small number of pregnant women are meeting recommendations for intakes of these food groups, this may suggest that intake of dairy and beef for non-pregnant women may be problematic as well. This contradicts findings from the study by Blumfield et al., where there were no significant differences between the intakes of dairy and meat between pregnant and non-pregnant women trying to conceive, respectively [1]. However, this may not factor the higher pregnancy NRVs, which are not reflected as more nutrient dense diets in pregnant women verses non-pregnant women [1].

Consistent with previous findings, the recommended servings of the breads and cereals group was met by the least number of women (*n* = 3, 0.6%). Studies by Lee et al. and Malek et al., have reported 1.8% and 4% of pregnant women respectively meeting the 2013 AGHE for breads and cereals [7,12]. The lack of consumption of grains is consistent with low fibre intakes, where 211 (39.5%) of participants in the current study met the NRV of 28 g of fibre per day. The low grain intake observed also resulted in suboptimal carbohydrate intake, where the median intake of carbohydrates was on the low end of the AMDR at 45% of overall energy intake. However, low grain intakes may be associated with recruitment from high risk Gestational Diabetes clinics as patients were all on insulin and seeing a Dietitian once a fortnight. Low fibre intake can increase the likelihood of constipation, a problem commonly reported by women during pregnancy [7]. These results are consistent with previous studies and highlight the question as to whether the AGHE is achievable for pregnant women. Additional modelling may be required to inform contemporary and achievable diet recommendations. Alternatively, there may be a need for Australian practitioners to provide more support mechanisms to help pregnant women achieve these guidelines. The challenge is to optimise macronutrient intakes, given the trade-off between total carbohydrate and fat intake, and to avoid reliance of foods high in sodium [38]. The majority of women in the current study were exceeding the upper end of the recommended sodium intake (460–920 mg/day) and had a high total fat and saturated fat intake. There are potential health consequences associated with a high fat, high sodium diet, particularly in highly processed foods during pregnancy, including altered placental function and predisposition to metabolic disease in the offspring [38].

Only seven women met the pregnancy NRV for iron (22 mg/day) from food intake alone. This is consistent with the AGHE Food Modelling evidence, where all dietary models for Foundation and Total diets were unable to provide sufficient iron to meet the EAR in pregnant women [27]. The document suggests that iron supplements may be necessary and are commonly prescribed during pregnancy; however, it does not formally recommend iron supplementation in pregnancy [27]. A Canadian 2017 cross-sectional study in Quebec demonstrated that 97% of pregnant women had dietary intakes of iron below the proposed EAR; however, with supplementation, only 10% of women had inadequate iron intakes [39]. Iron deficiency during pregnancy poses serious health problems for the offspring and mother, such as preterm delivery, low birthweight and maternal depression [40,41]. The prevalence of postpartum iron-deficiency anaemia varies between 4–27% worldwide [42]. In a recent US cross-sectional study, in a sample of 102 non-anaemic pregnant women, 42% were observed to be deficient in iron [43]. As such, it is likely that the pregnancy iron NRV cannot be met by food alone, which reinforces the need for supplementing the diet with additional nutrients, for example through the fortification of cereal products and the recommendation of a prenatal or iron supplement.

Only four women (0.75%) met the NRVs for all five key nutrients important in pregnancy (fibre, calcium, iron, folate and zinc). These data indicate a low conformance with NRVs amongst the population. A systematic review by Hillier et al. stated that there is emerging evidence that women should have healthier eating patterns prior to pregnancy, although there is a lack of knowledge about the importance of nutritional intake during pregnancy [31]. However, they may not be aware of the need to increase important nutrients [36]. Aside from iron, calcium and fibre also had low adherence (40.3% and 39.5%, respectively). Additionally, folic acid, a nutrient widely known for its role in the prevention of neural tube defects, only had a 50% adherence rate [44]. There is a low proportion of pregnant women meeting NRVs, potentially due to impractical targets or a need for widespread dietary improvement amongst the population.

Results from the current study indicate the importance of additional vitamin and mineral supplementation during pregnancy, particularly for iron. The NRV for iron was met by the least number of women, and with iron removed from the analysis, 118 women met the NRVs for folate, calcium, zinc and fibre. Supplements were consistently used by 74% of women; however, we cannot determine their micronutrient contribution, due to lack of detail reported on brand and dosage. For a supplement to provide the EAR value of iron, it would need to provide approximately 12 mg, this value being the difference between the median iron intake in this analysis (10.1 mg) and the EAR (22 mg). For the context of this study, the aim was to determine whether pregnant women are meeting AGHE and NRV targets from food intake alone. However, understanding type and amount of supplementation routinely used by pregnant women is important in determining their micronutrient contribution during pregnancy. This is necessary to guide future revisions of dietary recommendations, as well as the advice from health professionals. Health professionals need to tailor diet and supplementation advice for iron to patients, to prevent intakes exceeding the upper limit of 45 mg per day. Higher dose supplementation with or without sufficient dietary iron can result in unpleasant side effects in the gastrointestinal tract, such as constipation, nausea and vomiting [45]. Median intakes were above their respective EAR values, at 525.75 µg (vs. 520 µg) for folate and 10.76 mg (vs. 9.0 mg) for zinc. A US cross-sectional study, comparing the usual dietary intakes of 1003 pregnant women, observed the mean folate intake from food alone at 630 µg/day vs. 1451 µg/day with supplementation [46]. Whilst adequate folic acid is widely known for its role in the prevention of birth defects, health professionals need to ensure their recommendations do not lead women to exceed the upper limits of these nutrients during pregnancy [46]. Supplementation in pregnancy has shown to be an effective and cost-effective mechanism for improving maternal nutrient intake compliance with NRVs and reducing the risk of adverse health outcomes in infants [47]. However, the changes in maternal hormones lead to adaptions in the utilisation of maternal nutrients, in order to ensure the foetus receives a continual supply of nutrients for growth and development [48]. These adaptive responses support women in meeting increased demands for nutrients despite the nutritional intake and status of the mother [48].

The food group intakes of the four women who met the NRVs for fibre, calcium, iron, folate and zinc differed to those recommended in the AGHE. These findings, although a small sample, are consistent with a 2011 study by Blumfield et al., where it has been shown that women can achieve nutrient targets without adhering to food group targets [1]. The three women in this subgroup who did not misreport had higher conformance to the AGHE than demonstrated by the entire cohort. Nevertheless, no member of this sub-group met all of the AGHE recommendations.

The findings from the current study suggest that the AGHE and NRVs may not take into account current dietary intake patterns. More diversity in food group recommendations that better align with the eating patterns of Australian pregnant women is needed. Barriers to healthy eating during pregnancy, such as food cravings, morning sickness and constipation, may contribute to non-conformance with the nutrition guidelines [49]. The lack of knowledge of the contents, or even the existence of these nutrition guidelines may also play a role [49]. Identifying the areas of the maternal diet that differ to national recommendations and understanding these dietary patterns can assist in informing future revisions of guidelines and targeting nutrition interventions accordingly. Additionally, understanding the dietary intake patterns of women preconception and postpartum will allow interventions to target women of childbearing age, which may be an effective way to improve nutrient intake and ultimately pregnancy outcomes. Data that uses current eating patterns should be considered in future modelling and food selection guide revisions, to develop nutrition guidelines that women of reproductive age can follow. Perhaps there is a need to analyse the diet using more than one dietary assessment method, to limit the respective biases associated with each methodology. Additionally, further examination into supplementation, together with food, will be highly beneficial as evidence for nutrition recommendations. Future research should observe dietary and lifestyle behaviours, food group intake and supplementation, to provide health professionals with the knowledge and evidence to deliver dietary advice.

A strength of the current study is that it uses dietary data collected post 2014 to compare intakes of Australian pregnant women to the current AGHE. A limitation is that it only captures data from one urban area in NSW, albeit using a large sample size, the John Hunter Hospital being the main referral hospital for the Hunter area. However, the data collection did not collect demographic data on other ethnicities, which is a potential limitation in terms of understanding dietary intakes in relation to recommendations. The John Hunter Hospital antenatal outpatient clinic services both medium- and high-risk patients requiring ongoing management of Gestational Diabetes Mellitus, pre-eclampsia or those who have had previous adverse outcomes, women with babies in breech position or those who are attending the clinic drug and alcohol services or Indigenous health services. As such, the dietary intake of participants may be influenced by socio-economic status or a medium- to high-risk health conditions. For example, carbohydrate and fibre intake of participants may be influenced by the presence of gestational diabetes. This study uses the AES FFQ, previously shown to be valid and reliable in assessing usual intake up to six months; however, it has not specifically been validated for pregnant women. Data was captured during the 3rd trimester only (28–36 weeks’ gestation) and does not take into account differences in intake over the trimesters. However, previous studies indicate little change in dietary patterns over the course of the pregnancy [50,51]. FFQs have a low participant burden compared to other forms of dietary assessment, although self-reported dietary data poses the risk of misreporting. This limitation has been addressed using energy cut off points by Meltzer et al. [29]. Further, the AES FFQ, although previously used in pregnancy [51], has not been validated in this population group, and therefore, the findings of this study need to be interpreted in this context. The current study also examined nutrient intakes from dietary data alone, noting that 74% of women were taking supplements. Due to inconclusive data reporting on supplement usage and branding, the analysis could not determine the micronutrient contribution from these supplements. As a result, only a small sample of women met the NRVs for iron, folate, calcium, fibre and zinc from food intake alone. However, the aim was to determine whether pregnant women are meeting AGHE and NRV targets from food intake alone.

## 5. Conclusions

Dietary patterns indicate that in this sample of pregnant women attending a major public antenatal hospital, intakes are not aligned with national recommendations for pregnancy. This highlights that pregnant women need more support to improve their dietary patterns in order to optimise micronutrient intakes. Those who met pregnancy specific NRVs had dietary patterns more closely aligned to AGHE targets, although they still did not meet these recommendations. This suggests either that women may not be aware of the guidelines or that targets are not achievable. Ideally, all women should be provided with evidence-based nutrition advice during pregnancy. There is a need to raise awareness among antenatal healthcare providers of the low adherence to national recommendations and ensure that all women receive accurate information about dietary intake and vitamin and mineral supplementation, as a strategy to optimise maternal and infant health. In addition, future revisions of the AGHE should take into account the current eating patterns of pregnant women and consider the recommendation of supplementation for those who are unable to meet targets.

## Figures and Tables

**Table 1 nutrients-12-02438-t001:** Socio-demographic characteristics of participating pregnant women (*n* = 534).

Variables	Value	
*n* = 534, unless otherwise stated ^†^	Mean	CI
Age (years)	30.0	[29.5–30.4]
Gestation period (weeks) at time of survey *n* = 515	31.4	[31.2–31.7]
	n (%)	
Married/de facto status	469 (87.8)	
Aboriginal or Torres Strait Islander	34 (6.4)	
Born in Australia	481 (89.9)	
English is the only language spoken	504 (94.4)	
Above high school qualification *	350 (65.5)	
Difficulty managing available income ^¶^	238 (44.6)	
Current smoker (*n* = 532)	38 (7.1)	
Current supplement user	395 (74.0)	
BMI (kg/m2) ^‡^ (*n* = 446)		
Underweight (<18.5)	20 (4.5)	
Normal (18.5–24.99)	171 (38.3)	
Overweight (25–29.99)	80 (17.9)	
Obese (30)	176 (39.2)	
In the past 12 months, the individual/family ran out of food or could not afford to buy more (*n* = 533)	21 (3.9)	
First pregnancy	202 (38.0)	

CI, confidence interval. ^‡^ BMI, body mass index. BMI determined from self-reported height and weight. * Above high school qualification = trade or apprenticeship, certificate or diploma, university degree and higher university degree. ^¶^ Difficulty managing available income included “It is impossible”, “It is difficult all of the time” and “It is difficult some of the time”. ^†^ Not all questions were forced, and thus, numbers who provided information on the socio-demographic characteristics vary.

**Table 2 nutrients-12-02438-t002:** Daily food consumption in pregnant women from the John Hunter Hospital antenatal clinic.

		All Women (*n* = 534)		Excluding Energy Intake Mis-Reporters (*n* = 503) ^¶^	
		Median	IQR	Median	IQR
**Food Groups**					
**Food Group Servings (Servings/Day) ***	**AGHE (Serves/Day)**				
Breads and Cereals	8.5	2.7	1.7–3.7	2.8	1.8–3.7
Fruit	2	1.7	1.0–2.6	1.8	1.0–2.6
Vegetables and Legumes	5	3.8	2.8–5.1	3.9	2.9–5.2
Dairy	2.5	1.3	1–1.8	1.4	1.0–1.9
Meat and Alternatives	3.5	2.3	1.6–3.3	2.4	1.7–3.3
**Nutrient Intakes**					
	**NRVs (Unit/Day)**				
**Macronutrients**					
Energy (kJ) with Dietary Fibre	-	8079	6468–9966	8280	6718–10,004
CHO (% E)	AMDR 45–65%	45	41–49	45	41–49
Protein (% E)	AMDR 15–25%	18	16–20	18	16–20
Fat (% E)	AMDR 20–35%	37	34–40	37	34–40
Sat. Fat (% E)	<10%	14	13–16	14	13–16
Omega 3 (mg)	-	170.2	109–257	179.4	119–263
Fibre (g)	28	25.8	19–33	26.3	20–33
% Energy from Core Foods	-	67	58–75	68	59–75
% Energy from Non-Core Foods	-	33	25–42	32	25–41
**Micronutrients**					
Thiamin (mg)	EAR 1.2	1.5	1.1–2.0	1.5	1.1–2.0
Riboflavin (mg)	EAR 1.2	2.0	1.5–2.5	2.0	1.5–2.5
Niacin Equivalents (mg)	EAR 14	35.8	28.4–45.0	36.8	30.0–45.6
Vitamin C (mg)	EAR 40	162.8	115.1–220.4	166.2	119.5–223.0
Dietary Folate Equivalents (µg)	EAR 520	525.8	406.3–668.3	537.4	427.0–670.2
Retinol Equivalents (µg)	EAR 550	903.3	632.3–1195.3	920.9	674.7–1209.1
Magnesium (mg) 19–30 Years Old	EAR 290	359.4	293.6–446.1	366.4	303.3–447.7
Phosphorus (mg)	EAR 580	1345.4	1059.0–1665.5	1364.3	1103.0–1674.9
Calcium (mg)	EAR 840	769.5	557.5–968.5	785.0	583.4–974.2
Iron (mg)	EAR 22	10.1	7.9–12.6	10.3	8.2–12.7
Zinc (mg)	EAR 9.0	10.8	8.6–13.5	11.0	8.9–13.6
Sodium (mg)	AI 460–920	1733.7	1325.6–2190.4	1775.9	1382.8–2203.3
Iodine (ug)	EAR 160	121.8	88.3–156.4	123.4	95.1–157.9
Potassium (mg)	AI 2800	3152.5	2531.8–3892.2	3199.0	2623.2–3947.5

IQR, interquartile range; EAR, estimated average requirement; AI, average intake. ^¶^ Determined those who likely misreported using Meltzer et al., cut off values (<4.2 mJ or >20.0 mJ/day) [29]. * Serving size: (a) Breads and cereals: Bread 40 g, cereal 30 g, cooked porridge 120 g, muesli 30 g, cooked rice/pasta/noodles/barley/quinoa 70–120g, dry biscuits 40 g; (b) Fruit: Whole fruit (including canned) 150 g, fruit juice 125 mL, dried fruit 30 g; (c) Vegetables: Cooked or fresh vegetables 75 g; (d) Dairy and alternatives: Milk 250 mL, hard cheese 40 g, soft cheese (ricotta) 120 g, yogurt 200 g; (e) Meat and alternatives: Lean (cooked) beef/veal/lamb/pork/65 g, poultry (cooked) 80 g, fish (cooked)100 g, eggs 120 g, nuts/seeds/nut butters 30 g, tofu 170 g, cooked or canned legumes 150 g; (f) Extras: Sweet biscuit 35 g, sweet pastries/cakes/pies 40 g, savoury pies/pastries 60 g, pizza 60 g, hamburger 60 g, chocolate 35 g, processed meats 110 g, sausage 50–60 g, potato crisps/corn chips 30 g, jam/honey 45 g, ice-cream 75 g, fat spread 20 g, sugar 40 g, light beer 600 mL, full strength beer 400 mL, wine (including sparkling) 200 mL, spirits/liqueurs 60 mL, fortified wine 60 mL.

**Table 3 nutrients-12-02438-t003:** Percentage of pregnant women achieving Australian Guide to Healthy Eating (AGHE) daily food group recommendations.

		All (*n* = 534)		Excluding Energy Intake Mis-Reporters (*n* = 503)	
Meeting Recommendations *	AGHE (Serves/Day)	n	%	n	%
Breads and cereals	8.5	3	0.6	3	0.6
Fruit	2	204	38.2	202	40.2
Vegetables, including legumes	5	142	26.6	139	27.6
Dairy	2.5	72	13.5	70	13.9
Meat and alternatives	3.5	97	18.2	95	18.9

* Defined by the Australian Guide to Healthy Eating food group recommendations for pregnant women.

**Table 4 nutrients-12-02438-t004:** Estimated proportion of pregnant women whose intakes met Nutrient Reference Values (NRVs).

		All (*n* = 534)		Excluding Energy Intake Mis-Reporters (*n* = 503)	
	NRVs (Units/Day)	n	%	n	%
**Macronutrients**					
CHO (% E)	AMDR 45–65%	287	53.7	272	54.1
Protein (% E)	AMDR 15–25%	455	85.2	433	86.1
Fat (% E)	AMDR 20–35%	177	33.1	165	32.8
Sat. Fat (% E)	<10%	39	7.3	33	6.6
Fibre (g)	28	211	39.5	208	41.4
**Micronutrients**					
Thiamin (mg)	EAR 1.2	363	68	356	70.8
Riboflavin (mg)	EAR 1.2	455	85.2	448	89.1
Niacin Equivalents	EAR 14	523	97.9	502	99.8
Vitamin C (mg)	EAR 40	526	98.5	501	99.6
Dietary folate equivalents (µg)	EAR 520	272	50.9	270	53.7
Retinol Equivalents (µg)	EAR 550	440	82.4	429	85.3
Magnesium (mg)	EAR 290	410	76.8	406	80.7
	EAR 300	387	72.5	383	76.1
Phosphorus	EAR 580	516	96.6	501	99.6
Calcium (mg)	EAR 840	215	40.3	212	42.1
Iron (mg)	EAR 22	7	1.3	5	1.0
Zinc (mg)	EAR 9.0	378	70.8	374	74.4
Sodium (mg)	AI 460–920	32	6	13	2.6
Iodine (µg)	EAR 160	121	22.7	119	23.7
Potassium (mg)	AI 2800	349	65.4	346	68.8
Meeting key pregnancy nutrients iron, folate, zinc, calcium and dietary fibre	-	4	0.75	3	0.56

EAR, estimated energy requirement; AI, average intake.

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
