# Peer review of "Do the Dietary Intakes of Pregnant Women Attending Public Hospital Antenatal Clinics Align with Australian Guide to Healthy Eating Recommendations?"

_nutrients, 2020, doi:10.3390/nu12082438_

Round 1
Reviewer 1 Report
Slater et al present a manuscript detailing dietary intakes in pregnant women in an Australian population. They report that few women fully adhere to the nutritional guidelines. This study is informative and well written. The study is certainly important in relation to public health and information of this nature is essential for promoting health messages. I have a few minor queries and comments.
The focus is obviously maternal diet but it would be good to have some information in relation to the general population as a reference. Ie is it the case just during pregnancy or is this a systemic problem in the population at large? Is there any evidence that diet quality worsens during pregnancy?
I understand that the recommendations are there to meet specific dietary guidelines but the volume of food required to meet some of the guidelines seems excessive, do the authors think that the guidelines are not being met for this reason? So are the women eating a relatively healthy diet and just not achieving the guidelines in terms the absolute amount of serves or is there an issue with poor quality diet?
Did the FFQ account for foods relevant to a range of cultural backgrounds?
Could the authors please provide a consort diagram for their study.
The lack of information around supplementation is a flaw in the study. I understand that the study itself is designed to assess whether the diet is adequate and meets guidelines but it would be really useful to know what type of supplements and %women are taking supplements, this should be documented in the discussion.
In relation to the food tolerance/allergy not being assessed, while this may not be relevant for some categories it would surely be important for meeting dairy guidelines in those that are intolerance/allergy to dairy, particularly in a multi-cultural society like Australia where there is a large proportion of individuals who may be lactose intolerant?
The demographic information only provides for Aboriginal/Torres Straight Islander and individuals born in Australia. Was the ethnicity broken down any further?
Are the BMI categories consistent with the general trend in Australia?
Reviewer 2 Report
Thank you for the opportunity to review the paper. I believe it is a well-written paper and adds to the literature on whether pregnant women's diets meet guidelines for food group consumption. I have some minor comments.
- Nearly 60% of the women in the study had pre-pregnancy overweight/obesity. It would be good to stratify the findings by weight status as food consumption will change by weight status.
- Did the authors measure physical activity and sedentary activity level of women? That would be a good addition to the paper to see how food consumption differed in meeting guidelines by activity status.
- The authors mention a number of comorbidities that would influence daily lifestyle, including gestational diabetes, gestational hypertension, prenatal depression, etc. It would be beneficial to conduct the analysis limiting to pregnant women without these risk factors. Same goes for limiting the analysis to women who have difficulty managing available income (44.6%).
- I agree with the authors that women may not generally be aware of food and nutrient consumption guidelines during pregnancy.
- It is a limitation that authors could not ascertain how much of the micronutrient was coming from supplements. A majority of pregnant women would be on prenatal vitamins, folic acids, and iron during pregnancy.
- Another interesting study might be to see how dietary habit changes during postpartum and whether women meet guidelines for food and nutrient consumption.
